# Occurrence of Microplastics in Waste Sludge of Wastewater Treatment Plants: Comparison between Membrane Bioreactor (MBR) and Conventional Activated Sludge (CAS) Technologies

**DOI:** 10.3390/membranes12040371

**Published:** 2022-03-29

**Authors:** Gaetano Di Bella, Santo Fabio Corsino, Federica De Marines, Francesco Lopresti, Vincenzo La Carrubba, Michele Torregrossa, Gaspare Viviani

**Affiliations:** 1Faculty of Engineering and Architecture, University of Enna “Kore”, 94100 Enna, Italy; 2Department of Engineering, University of Palermo, 90128 Palermo, Italy; santofabio.corsino@unipa.it (S.F.C.); federica.demarines@unipa.it (F.D.M.); franceco.lopresti01@unipa.it (F.L.); vincenzo.lacarrubba@unipa.it (V.L.C.); michele.torregrossa@unipa.it (M.T.); gaspare.viviani@unipa.it (G.V.)

**Keywords:** membrane bioreactors, microplastics, waste sludge, wastewater treatment plant

## Abstract

In this study, the presence of microplastics in the sludge of three wastewater treatment plants (WWTPs) was examined. The investigated WWTPs operated based on a conventional activated sludge (CAS) process, with (W1) or without (W2) primary clarification, and a membrane bioreactor process (MBR) (W3). The microplastics (MPs) concentration in the samples of W3 was approximately 81.1 ± 4.2 × 10^3^ particles/kg dry sludge, whereas MPs concentrations in W1 and W2 were 46.0 ± 14.8 × 10^3^ particles/kg dry sludge and 36.0 ± 5.2 × 10^3^ particles/kg dry sludge, respectively. Moreover, MPs mainly consisted of fragments (66–68%) in the CAS plants, whereas the fractions of MPs shapes in the MBR sludge were more evenly distributed, although fiber (47%) was the most abundant fraction. Furthermore, samples from the MBR showed a greater diversity in MPs composition. Indeed, all the main polyesters (i.e., textile fibers and polyethylene terephthalate), polyolefins (i.e., polyethylene and polypropylene) and rubber (i.e., polybutadiene) were observed, whereas only polybutadiene, cellulose acetate and polyester were detected in the CAS plants. These findings confirmed that MPs from wastewater are transferred and concentrated in the waste sludge. This is a critical finding since sludge disposal could become a new pathway for microplastic release into the environment and because MPs might affect the fouling behavior of the membrane.

## 1. Introduction

During the last decade, microplastics (MPs) pollution has become an emerging problem for the environment. Because of their potential to absorb organic contaminants, MPs are considered emerging pollutants of public health and environmental concern [1]. Indeed, MPs degrade very little and accumulate in the oceans, causing eco-toxicological issues in aquatic environments and ecosystems [2]. Microplastics are defined as particles of synthetic origin having dimensions smaller than 5 mm. Generally, they are divided into primary microplastics and secondary microplastics [3]. The former refers to plastic products having a microscale size at their origin, whereas the latter are generated from weathering and abrasion of large plastic particles [4]. Therefore, MPs could be derived from a wide variety of sources, including synthetic fibers from clothes, personal care products, polymer processing industries, etc.

Wastewater treatment plants (WWTPs) are noted to be both sinks and sources of MPs [2]. Several studies reported the occurrence of MPs in both WWTP influent and effluent wastewaters as a result of daily human life activities [5]. For example, polyester and polyamide fibers derived from laundry activities, personal care products from daily use, rubber from road washout, etc. [6]. According to previous studies, approximately 80–90% of MPs entering a WWTP are removed after mechanical, chemical, and biological treatment [3]. The removal efficiency of MPs depends on the WWTP configuration. Indeed, in a typical conventional activated sludge (CAS) based WWTP, the concentration of MPs after primary treatments is reduced by up to 70%, whereas it can be further reduced by 80–90% and 99% in the secondary and tertiary effluents, respectively [7]. In particular, filter-based technologies, such as biofilters, ultrafiltration (UF) and rapid sand filters (RSFs), achieve the best performance in removing microplastics. Primary settling treatment is highly efficient in removing microplastics, while grit and grease treatment exhibit a low efficiency in microplastics removal. The removal efficiency of the biological process is closely related to the characteristics of wastewater and the types of microplastic polymers [5].

Although high removal efficiencies are achievable, the effluent and sludge of wastewater usually contain high concentrations of microplastics. For example, studies have shown that the range of abundance in the sludge of WWTPs is 1000–24,000 n⋅kg [8]. Moreover, previous studies demonstrated that more than 80% of plastic particles entering WWTPs are retained in the sludge produced during wastewater treatment processes. This could result in a further pathway for the release of MPs into the environment, depending on the sludge disposal method adopted [9]. For instance, it was found that the rate of deposition of microplastics in the soils of Norway has reached five hundred billion tons each year, while in the soils of North America the figure is 44,000–430,000 tons per year [10].

In recent years, membrane bioreactor technology (MBR) has been widely used for wastewater treatment due to its potential to produce high quality effluent and remove emerging micropollutants, while reducing the plant footprint [11]. In this context, several studies evaluated the combination of MBR with other existing processes to reach a higher effective removal of microplastics from wastewater [12]. A recent study reported that the MBR process had a better removal efficiency of MPs compared with a CAS-based process [13]. Similarly, in another study a research group found that MBR produced the highest percentage removal of MPs (99.9%) compared with CAS processes integrated with tertiary treatment technologies (i.e., disk filter, rapid sand filter, dissolved air floatation) [14]. Nevertheless, several papers demonstrated that the fraction of MPs removed from wastewater is trapped in the sludge phase [15,16]. Thus, as previously stated, sludge disposal (landfilling, incineration, or land application), becomes a new pathway for microplastic release into the environment. Therefore, a careful examination of the occurrence of microplastic in the waste sludge as the most important recipient of such compounds is very important. Most of the available studies have investigated the occurrence of MPs in the sludge of CAS-based WWTP as a function of various technologies applied in the wastewater handling units [5]. Nevertheless, to our knowledge, no evidence of the higher abundance of MPs in the waste sludge from WWTP based on the MBR process is reported in the literature.

In this light, the aim of this study was to investigate the occurrence of MPs in the waste sludge from an MBR-based WWTP. Specifically, the objectives of this study were: (1) to analyze the concentration of MPs in the waste sludge of three different WWTPs, two based on a CAS process and one on MBR technology, and (2) assess the difference in shape and composition of MPs in the waste sludge samples.

## 2. Materials and Methods

### 2.1. Description of WWTPs

The selected WWTPs were all full-scale plants treating domestic wastewater, located in Sicily in the south of Italy. All the WWTPs were served by combined sewer systems. The first WWTP, named W1, was configured according to a classical treatment scheme including a screening with step screen of 20 mm, grit separation, primary clarification, biological treatment with activated sludge, final sedimentation and disinfection in the water line, whereas aerobic digestion, sludge thickening and dewatering through a filter press were conducted in the sludge line. The second WWTP, named W2, was similar to W1, apart from the primary clarification in the water line and a centrifuge for the sludge dewatering process. Finally, the third WWTP implemented an MBR process, consisting of screening with a step screen of 20 mm followed by a micro-sieving unit (3 mm), grit separation and a membrane bioreactor in the water line, whereas aerobic digestion, sludge thickening and dewatering through a centrifuge occurred in the sludge line. The W3 was equipped with ultrafiltration hollow-fiber membranes in a submerged configuration having a porosity of 0.04 µm. W1 and W3 were located near the coastal area, whereas W2 was inland. The population served was approximately 330,000 PE, 12,000 PE and 27,000 PE, for W1, W2 and W3, respectively. The plant capacity of W1 was close to 86,200 m^3^/d, whereas that of W2 and W3 were 3000 m^3^/d and 6000 m^3^/d, respectively. Figure 1 summarizes the layout of the three WWTPs.

### 2.2. Samples Collection

Sludge samples were collected from each of WWTPs during the spring/summer period, more precisely between April and June 2021. This was because in a previous study it was demonstrated that the maximum number of MPs entering a WWTP was observed during the spring/summer period, according to people’s habits and tourism activities [9]. Specifically, samples were collected from all the WWTPs on the same day, once a month, over the three-month period. Each sample was collected after the dewatering unit. Approximately 10 kg of dewatered sludge was collected from each plant and stored at 4 °C in a stainless-steel bucket to prevent the introduction of plastics prior to analysis. The total solid content of the sludge samples was determined by drying the samples at 105 °C for 24 h. The total solid content of the sludge samples ranged from 20% to 26%.

### 2.3. Samples Treatment

Before the extraction, all sludge samples were carefully homogenized. The MPs were extracted from the sludge samples using a multiple-step procedure derived from a previous study [17]. Samples were dried overnight in a ventilated oven at 60 °C. Then, the dried sludge samples were gently ground and sieved through a 5 mm and a 1 mm stainless steel mesh. For each of these fractions (<1 mm, 1–5 mm), approximately 10 g of dried sludge was added to a 500-mL conical flask with approximately 200 mL of 30% H_2_O_2_ solution and heated for 3–4 h on a digestion plate at 70 °C until the solution was nearly dried. After the samples had cooled, approximately 200 mL of a floatation solution (350 g NaCl/L) was added to the conical flask. The flask was covered and mixed for approximately 1 h using a magnetic stirrer plate. After mixing, the samples were left to stand for 2 days. Thus, approximately 100 mL of supernatant was removed and replaced with the same volume of floatation solution. This procedure was repeated twice, and the supernatants collected at the end of each floatation phase were combined. The MPs in the supernatant were then separated by vacuum filtration using a 20 µm nylon membrane. The membrane was carefully washed with deionized water to remove the salt from the MPs. A post-digestion treatment was applied to remove residual organic materials. Specifically, the MPs were washed first with 30% H_2_O_2_ and after with 25% H_2_SO_4_. Then, the MPs were carefully rinsed with Milli-Q water to remove the residual traces of H_2_O_2_ and H_2_SO_4_. The MPs were placed in a 40-mm Petri dish and residual natural debris was removed with a stainless-steel tweezer.

### 2.4. Examination and Identification of MPs

A visual identification was performed to classify the extracted MPs. Specifically, MPs were examined under a digital stereomicroscope (Olympus SZX7, Olympus Co., Tokyo, Japan) with a magnification up to 90×. The stereomicroscope was coupled to a Leica HD digital camera and the image capturing software Leica (LM50). The extracted particles were evenly distributed on a 40-mm Petri dish and overlaid with a square grid mesh (1 cm × 1 cm). During the observation process, the filter was moved under the stereomicroscope up and down in a “U” pattern until all the particles within a 1 cm^2^ of mesh were photographed. For each sample, all the particles within at least nine mesh grid squares were entirely detected. Then, the particle density (number/cm^2^) was calculated in reference to the entire surface area of the Petri-dish. Therefore, the concentration of microplastic was expressed as the number of particles in reference to the unit (kg) of dry sludge. Once the images were captured, MPs were measured and classified according to color (transparent, white, red, black, blue, brown and green) and type (fiber, film, fragments and spheres).

Fourier transform infrared spectroscopy in attenuated total reflection (FTIR-ATR) analysis was performed to confirm the plastic or non-plastic nature of the extracted particles and to assess their molecular composition. Polymers were identified by means of different reference polymer libraries, containing spectra of all common polymers, such as polybutadiene (PB), cellulose acetate (CA), textile polyester (PL), polyethylene (PE), polypropylene (PP), polyethylene terephthalate, (PET), etc. The analysis was performed on a statistically significant number of MPs using a Perkin-Elmer FTIR-NIR Spectrum 400 spectrophotometer. The spectra were recorded in the range 4000–400 cm^−1^. On average, from 20 to 40 particles of each MP category were examined with FTIR, corresponding to 10% to 15% of all the collected particles.

## 3. Results and Discussion

### 3.1. Microplastic Concentration and Distribution

MPs were found in all the samples collected from the three WWTPs. Figure 2 shows the abundance of MPs with sizes smaller than 1 mm (Figure 2a) and between 1 and 5 mm (Figure 2b) detected in the sludge samples. Specifically, the abundance of MPs in the sludge samples of W3 was greater than that of the other two WWTPs. Indeed, the MPs concentration detected in the samples of W3 was approximately 81.1 ± 4.2 × 10^3^ particles/kg dry sludge, whereas the concentrations in W1 and W2 were 46.0 ± 14.8 × 10^3^ particles/kg dry sludge and 36.0 ± 5.2 × 10^3^ particles/kg dry sludge, respectively. Similar results were obtained for the particles with a size between 1 and 5 mm. In fact, a greater abundance of microplastics was found in the W3 samples, with an average of 1.1 ± 0.4 × 10^3^ particles/kg dry sludge, whereas the concentrations observed in the samples of W1 and W2 were on average 0.28 ± 0.14 × 10^3^ particles/kg dry sludge and 0.50 ± 0.53 × 10^3^ particles/kg dry sludge, respectively. Based on these results, the abundance of MPs was much greater in the WWTP with MBR technology. Indeed, the MPs concentration was almost double compared with that detected in the other two plants with CAS technology, suggesting that the higher removal efficiency of MPs from the wastewater observed in previous studies in MBR based WWTP resulted in the greater accumulation of such particles in the waste sludge [13,18]. Moreover, there was a greater standard deviation for the sludge samples from W1, suggesting greater variability in the occurrence of MPs in the sludge of the CAS plant with the primary settling unit. Previous studies reported that the abundance of MPs in the sludge of WWTP with primary settling is higher than those without primary clarification of wastewater [19]. However, in this case, it is possible that the greater variability in abundance of MPs observed in the sludge of W1 could be also attributed to the greater daily treatment capacity of this plant than the other two. Indeed, a previous study demonstrated that the amount of microplastics found in the sludge is proportional to the volume of wastewater treated [20]. Overall, the concentrations of MPs with sizes lower than 1 mm were significantly greater than that of size between 1–5 mm, comprising more than 98%. The lower abundance of larger size MPs in the waste sludge of all the three WWTPs could be attributed to the removal of these particles in the wastewater pre-treatment units (i.e., screening and grit removal). Several studies reported that a screen and aerated grit chamber could effectively retain large debris and suspended particles, leading to a significant reduction of MPs in wastewater and their accumulation in the screening residues and sand [21,22]. Some studies report that pre-treatment removes fibers more effectively than fragments, while secondary treatment removes more fragment particles than fibers [12]. These two wastes are disposed of separately from waste sludge, having a different classification code according to the European Waste Catalogue (EWC 19.08.01: screening residues; EWC 19.08.02: waste for the elimination of sand; EWC 19.08.05: sludge produced by urban wastewater treatment). Therefore, the lower amount of these particles in the waste sludge could be related to their accumulation in other waste upstream from the biological processes. Stereoscopic images showed that these particles were characterized by very irregular shapes, thereby suggesting that these particles were secondary MPs [8].

### 3.2. Morphology of MPs

The stereomicroscopic images reveal that the MPs were characterized by different morphologies. Figure 3 shows the percentage of the main types of shapes for MPs in the sludge of W1, W2 and W3. 

Fragments, films, and fibers were the dominant shapes of MPs observed in all the sludge samples of the three WWTPs. In contrast with the findings of other studies, no spheres were observed in this study. This could be attributed to the extraction method used in this study that likely caused the spheres’ deterioration during the acidification steps, although further investigations are necessary to clarify this aspect. MPs with sizes smaller than 1 mm in the samples of W1 and W2 mainly consisted of fragments (66–68%), whereas the percentage of films and fibers (13%, 8%) was significantly lower in both plants. In contrast, the MPs shape types in the W3 sludge were more evenly distributed. Indeed, the fibers in the sludge samples of W3 accounted for about 47% of the total, whereas films accounted for 29% and fragments for 24%. These results indicate that the shapes of the MPs from the two WWTPs with the CAS-based process were very similar and comparable with those reported in previous studies [6,8]. The reason for the greater abundance of fragments might be that these denser particles are more prone to form sludge sediments in the primary settling tanks. This result was also confirmed by a recent study, in which the authors demonstrated that the heavier microplastics are trapped in the primary sludge [7]. Similar results were obtained by Liu et al. who found fragments made up the largest proportion of microplastics identified in their sludge samples followed by fibers [23]. The greater percentage of films observed in the MBR sludge samples in comparison with the CAS samples is attributed to the retention effect exerted by the ultrafiltration membranes on these particles. Indeed, since film particles are bigger than the membrane porosity, they were entirely entrapped within the system and accumulated in the waste sludge. Films consist of low-density particles that are not able to settle and are not easily embedded within the activated sludge flocs in a CAS process. Therefore, these particles are washed out with the effluent wastewater in CAS systems [6,24]. In contrast, the retention of these particles is higher in the MBR, because gravity separation of sludge is replaced by a physical filtration, and the membrane’s physical barrier stops these particles. Similarly, the greater percentage of fibers observed in the MBR sludge samples is attributed to the higher retention capacity of ultrafiltration membranes toward these particles that are accumulated in the waste sludge. Indeed, previous studies carried out on CAS and MBR-based processes, demonstrated that fibers were the dominant shape for microplastics in the sludge of an MBR pilot plant, whereas their proportion in the CAS sludge was significantly lower and comparable with that obtained in the present study [13]. Similar results were also found by other research groups [20]. Fiber particles are not symmetrical and are characterized by having one dimension significantly longer than the other particle types. It is reported in the literature that the fraction of these particles in the sludge from conventional WWTPs is generally lower than the other microplastic shapes [21,25]. In a previous study, the fiber shape particles were highlighted as the most challenging MPs to be removed in WWTPs due to their smooth surface and high ratio of length to width [20]. Indeed, fibers were found as the main MPs in many effluent wastewaters indicating that these particles are not easily incorporated within the sludge flocs [19,26]. In contrast, the retention of these particles is higher in the MBR, because they cannot pass through the membrane pores. The lower percentage of fragments and film particles observed in the W3 samples could be related to the higher removal performances obtained with the pre-treatments in the MBR plant. Indeed, in MBR systems the pre-treatment processes are generally more advanced in order to prevent debris fouling and are aimed at removing such particles (i.e., fragments) that could be potentially dangerous for the membrane integrity [27]. In this case, the micro-sieving in the layout of W3 likely caused the decrease in the number of fragments and films passing through the pre-treatment units, leading to the lower percentage of these particles observed in the waste sludge of the MBR plant.

Referring to MPs with sizes between 1 and 5 mm, fragments (43%) and fiber (52%) were still the two main MP particles observed in the samples of W1 and W2, although the percentage of fibers significantly increased in these samples. Indeed, the fraction of fibers doubled in both the CAS-based WWTPs at the expense of the fragments, whose percentage decreased from 66% to 43% in W1 and from 68% to 37% in W2. The fraction of films bigger than 5 mm was the lowest in both the WWPTs (<10%). The greater percentage of fibers bigger than 5 mm in the CAS samples could suggest the higher tendency of the activated sludge to embed such particles into the flocs structure in comparison with the smaller fibers that tended to be washed out with the effluent wastewater.

The percentage of MP shapes in the W3 sludge was evenly distributed also in this case, suggesting that MBR can entrap all the particles within the sludge independent of their shapes. The fraction of fibers remained the most abundant accounting for 40%, whereas that of fragments and films was approximately 28% and 32%, respectively. This result was consistent with the results reported in a recent study conducted on an MBR-based WWTP [28]. Based on the above results, it could be stated that the sludge of the MBR presented a higher diversity of microplastic shapes, the dominant fraction of which was fibers. As is widely reported in the literature, debris fouling of hollow-fiber membranes is one of the main management issues of this technology, and fibrous materials are the main foulant agents in this context [29]. In general, this material mainly consists of hair, cellulosic fibers and textile fibers [30]. Previous studies demonstrated that such fibrous materials acted as a skeleton and contributed to a cake layer formation [31]. The removal of such material with ordinary backwashing is very difficult to achieve; thus these particles accumulate in the cake layer deposited on the membrane surface and contribute to the increase of its fouling mechanism [30]. The findings of this study suggested that the fibrous material that accumulates in the sludge of MBR systems also consists of fibers of synthetic origin that belong to the MPs category. Besides representing a severe environmental issue depending on the final disposal of waste sludge, this could be a potential stress factor for membrane fouling with consequent impacts on management costs.

### 3.3. MPs Composition

The analysis of microplastics composition could be a useful approach to evaluate their origin, thereby making it possible to limit their production directly at the source. MPs in this study were identified by FTIR-ATR analysis. Some of the FTIR-ATR spectra of the MPs extracted from the sludge samples of W1, W2 and W3 are depicted in Figure 4, and they were identified as polyethylene terephthalate (PET) [32], cellulose acetate (CA) [33], polybutadiene (PB) [34], and textile polyester (PL) [35].

Figure 5 reports the percentages of microplastics detected in the sludge samples of the three WWTPs.

Microplastic particles extracted from the sludge of W1 were mainly composed of PB, CA and textile PL, in accordance with other studies on the composition of MPs in sludge from conventional WWTPs [36]. PB and PE were the main MPs detected in the sludge, each accounting for approximately 35%, whereas cellulose acetate was close to 20%. In the sludge samples of W2, a lower diversity of microplastics was found in comparison with W1. Indeed, PB and PET were the most abundant polymers (55%) detected by FTIR analysis, whereas the presence of other compounds (PP, PE, PL) was secondary. In contrast to what was observed in previous literature, in this study, a significant amount of PB fragments was found in the sludge. These particles were black in color, indicating tires as a possible origin. These particles tend to deposit on the roadway, run off during rain events and thus arrive at the WWTPs [37]. The great abundance of these particles could be attributed to the sewer system. Indeed, the combined sewer system was associated with an increased number of these fragments in the wastewater. This could also be related to the adjacent surrounding land use as well as transport related emissions, such as microplastics released from the wear of tires and brakes [18].

In contrast, in the W3 samples, a greater diversity of plastics was observed. Indeed, all the main polymers such as PE, PL, PET, PP and PB were observed. The most abundant MPs were PET and PL (white, red and green) accounting for approximately 29% and 27%, respectively. PET was also the main MPs detected in another MBR-based WWTP [28]. The results above were consistent with the greater diversity of microplastic shapes observed in the MBR sludge as previously discussed. Therefore, MBR-WWTP showed a greater capacity to intercept more MPs of different shape and composition compared with the WWTPs with CAS technology. Moreover, the greater abundance of PET and PE was consistent with the higher number of fibers detected in the sludge of W3, since these microplastics originate from synthetic textiles (e.g., from t-shirts, polar fleeces and clothes) in the form of fibrous particles. Harley-Nyang et al. found that PE was the most abundant polymer detected in dewatered sludge samples [38]. In this case, the fraction of PB was lower compared to W1 and W2, in accordance with the lower percentage of fragments detected in the sludge of W3. This indicated that PB particles were likely removed in the pre-treatment units upstream from the biological process in the MBR-WWTP.

### 3.4. Comparison with Literature Data and General Remarks

It is quite difficult to compare the results of microplastics detected in the sludge samples with those reported in other studies. There are several reasons for this, including the different protocols applied for microplastics extraction, the differing capacities and layouts of the WWTPs and the specific characteristics of the single units they contain, the characteristics of the area served by the WWTPs, differing urban waste management practices, etc. Nevertheless, to highlight the greater abundance of MPs in the sludge of the MBR, a comparison was made with results reported in the literature on WWTPs similar to those examined in this study. Table 1 summarizes the main results obtained in studies carried out on municipal WWTPs with different treatment capacities.

The results obtained for the sludge of W1 and W2 were comparable with those reported in the main studies on MPs in waste sludge available in the literature. In general, in WWTPs implementing a primary clarification of wastewater and a CAS process, the number of MPs in the sludge is higher compared with a similar plant without the primary settling phase [21,39,40]. This is because pre-treatments have a high retention capacity for MPs, especially for large fragments and fibers. Consequently, in WWTPs without primary clarification, a greater number of microplastics exit from the plant in the effluent wastewater [6]. The number of MPs detected in the MBR was significantly higher than that reported in the CAS-based WWTPs. Moreover, the concentration of MPs detected in W3 was much higher than that observed in a CAS-WWTP with a treatment capacity much greater than the MBR examined in this study [41]. Overall, even the types of MPs found in the sludge of the MBR were different compared with those of the CAS WWTP. The greater abundance of fibers and the greater diversity of plastics found in the MBR sludge confirmed that MBR are more effective than CAS in removing MPs from wastewater. Nevertheless, the findings of this study also confirmed that microplastics removed from wastewater are transferred into the waste sludge. This is a critical concern since some sludge disposal practices could become new pathways for microplastics release into the environment. Moreover, further studies are necessary to investigate the effect of MPs on the hydraulic performance of the membrane in the long term since the presence of fibers could cause a significant worsening of the fouling problem. Therefore, greater attention should be paid to the effects of MPs on the treatment processes and the disposal pathways of waste sludge from MBRs.
membranes-12-00371-t001_Table 1Table 1Summary of the average abundance of MPs in the waste sludge from municipal WWTPs available in the literature in comparison with this study.Treatment in Water Line (Biological Phase)Potential (PE)LocationMPs (n/kg) *ReferenceCAS-based process (no primary settling)250 × 10^3^China44–750[8]Primary settling + CAS based process0.5–2.5 × 10^3^Ireland4.19–15.4 × 10^3^[40]CAS-based process (no primary settling)n.a.The Netherlands370–950[41]Primary settling + CAS-based process12–111 × 10^3^China1.65–56.3 × 10^3^[41]MBR (pilot plant)n.a.Finland27.3 × 10^3^[13]Primary settling + CAS-based process10–210 × 10^3^Germany1.0–24 × 10^3^[39]Primary settling + CAS-based process20 × 10^3^China24 × 10^3^[21]CAS-based process (no primary settling)48 × 10^3^Australia40 × 10^3^[38]Primary settling + CAS-based process45 × 10^3^Spain18.3 × 10^3^[42]CAS-based process (no primary settling)493 × 10^3^Canada14.9 × 10^3^[37]W1—Primary settling + CAS-based process330 × 10^3^Italy46 × 10^3^This studyW2—CAS-based process (no primary settling)12 × 10^3^Italy36 × 10^3^This studyW3—MBR (full-scale plant)27 × 10^3^Italy86 × 10^3^This study* data refer to dry sludge; n.a. = data not available.


## 4. Conclusions

In this study, the occurrence of MPs in the sludge of WWTPs with conventional and MBR technologies was examined. The findings of this work demonstrated that MPs present in the wastewater are accumulated in the waste sludge, thus representing an additional pathway for their release into the environment. Specifically, the MBR plant showed a significantly higher number of MPs in the sludge compared with the CAS-WWTPs. The overall abundance of MPs in the MBR was 81.1 × 10^3^ particles/kg dry sludge, almost double the amount of the other plants (36–46.0 × 10^3^ particles/kg dry sludge). Moreover, MPs in the MBR sludge exhibited a greater diversity of shapes (mainly fibers) and composition (PE, PET, PP, PB, etc.). This study provides useful information for better understanding the microplastics pollution in the sludge of conventional WWTPs and MBRs specifically. Further studies should be carried out to find suitable advanced techniques to remove MPs from the sludge before its final disposal. Lastly, the effect of MPs on membrane fouling behavior should be evaluated in future studies.

## Figures and Tables

**Figure 1 membranes-12-00371-f001:**
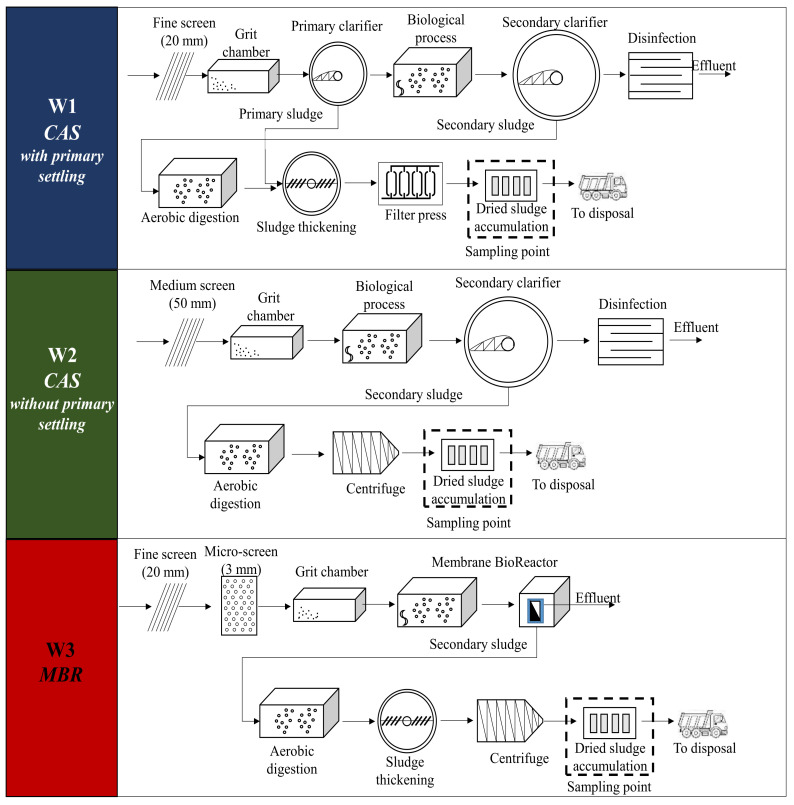
Layouts of the three WWTPs.

**Figure 2 membranes-12-00371-f002:**
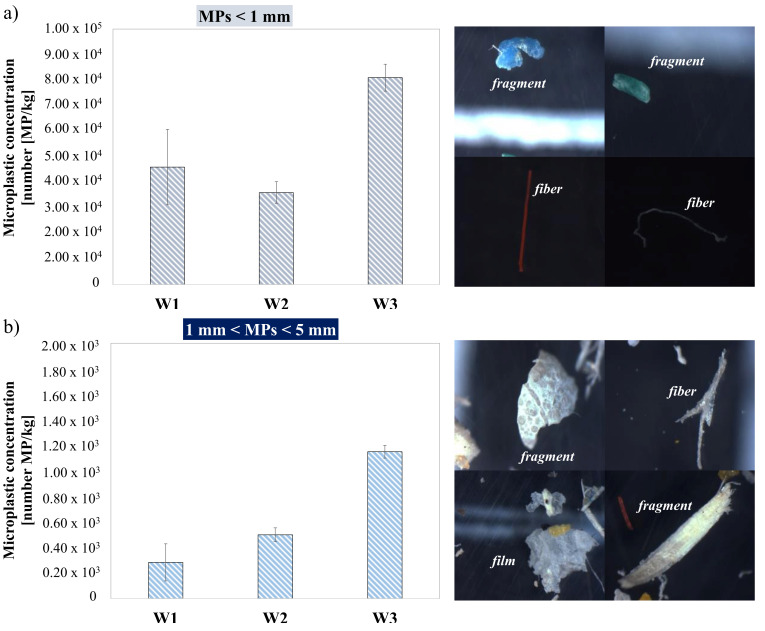
Average abundance of MPs in the dewatered sludge of W1, W2 and W3 with a size smaller than 1 mm (**a**) and between 1 mm and 5 mm (**b**); The pictures show stereoscopic images of MPs of different size and morphology detected in the W3 samples.

**Figure 3 membranes-12-00371-f003:**
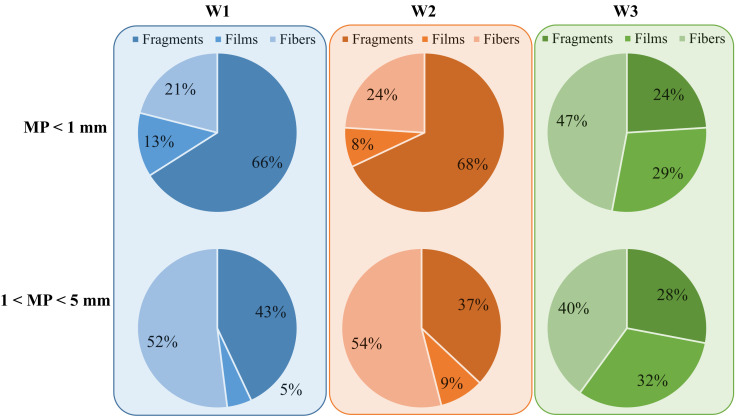
Distribution of MPs with different shapes in W1, W2 and W3.

**Figure 4 membranes-12-00371-f004:**
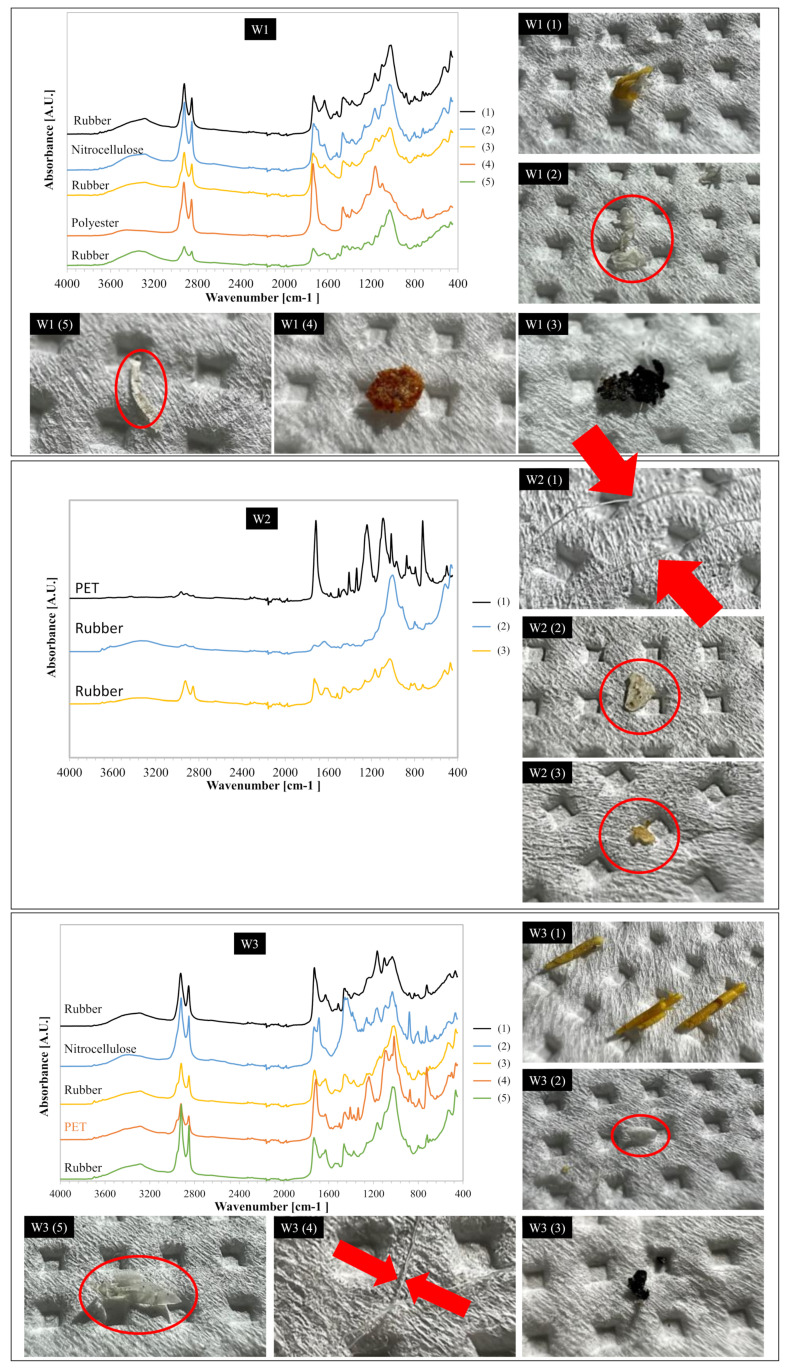
Representative ATR-FTIR spectra and images showing comparisons of some types of MPs found in the sludge samples of W1, W2 and W3.

**Figure 5 membranes-12-00371-f005:**
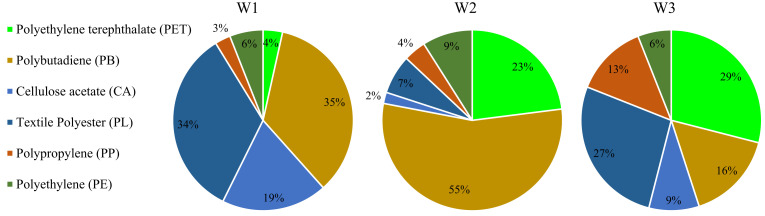
Average composition of MPs in sludge samples of W1, W2 and W3.

## Data Availability

Data will be available on request to the corresponding author.

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
