# Peer review of "Occurrence of Microplastics in Waste Sludge of Wastewater Treatment Plants: Comparison between Membrane Bioreactor (MBR) and Conventional Activated Sludge (CAS) Technologies"

_membranes, 2022, doi:10.3390/membranes12040371_

Round 1
Reviewer 1 Report
The study investigates the occurrence of microplastics in waste sludge from biological processes in the case of conventional activated sludge (with and without primary sedimentation) and MBR. The analysis refer to samples of dewatered sludge from 3 full different scale plants. (Different treatments are present for the sludge lines). Microplastics analyses refer to their size distribution and abundance in the different samples.
The manuscript is of great interest and deals with a very timely topic.
The main critical points I found in this paper is the description of the type of wastewater conveyed to the three WWTPs and the WWTPs themselves (capacity and flow rates do not well match). I think that further details on the type of wastewater treated could better support the discussion. In addition the comparison with other case studies is not sufficiently commented and requires a deeper analysis.
The paper after a moderate revision can be considered for publication as I think it will promote the discussion on these contaminants and further researches.
L53-54: If I well understand, the removal values provided refer to WWTP influent and primary effluent, or to secondary effluent or to tertiary effluent and not to the specific contribution of the step (primary, secondary and tertiary). I suggest to improve the clarity of the sentence.
L62-64: Ref 11 is a review and the results was not of the research group.
L82-97: I suggest to add if the sewer to the three WWTP is mixed or separate and the type of users served by the WWTP (domestic, industrial, zootechnic…). These details could be of interest to better compare the results achieved for the three WWTPs. For instance, at lines 292 authors related the occurrence of some particles with to tires and road runoff.
The only provided detail is that the three WWTPs treat domestic wastewater and are of different capacity. My doubt is how the differences in shape and composition can be due to the wastewater type treated at the specific plant?
L95: Is it correct that W1 serves 330.000 PE and it has a capacity of 3600 m3/d? Check served population and capacity!
Figure 2 enlarge axis titles and all the characters. It is not easy to evaluate the standard deviations in the bottom graph. As to pictures, it is not clear to which samples they refer to.
L209: which are shapes could be present but you did not find?
Figure 5: I recommend to add MP acronyms soon after their full name
L293-294: what does this mean? At line 104 Authors wrote three dewatered sludge samples were taken in the three WWTPs in the same days (in April, May and June 2021) and at line 294, they wrote samples were taken after rain event. Is this collection constraint necessary and within the scope of the current study?
General comments to figures: enlarge the characters.
Author Response
Reviewer 1
The study investigates the occurrence of microplastics in waste sludge from biological processes in the case of conventional activated sludge (with and without primary sedimentation) and MBR. The analysis refer to samples of dewatered sludge from 3 full different scale plants. (Different treatments are present for the sludge lines). Microplastics analyses refer to their size distribution and abundance in the different samples. The manuscript is of great interest and deals with a very timely topic.
The main critical points I found in this paper is the description of the type of wastewater conveyed to the three WWTPs and the WWTPs themselves (capacity and flow rates do not well match). I think that further details on the type of wastewater treated could better support the discussion. In addition the comparison with other case studies is not sufficiently commented and requires a deeper analysis.
The paper after a moderate revision can be considered for publication as I think it will promote the discussion on these contaminants and further researches.
Authors’ response: Authors thank Reviewer #1 for the interesting comments and suggestions aimed at improving the quality of the paper. As the Reviewer could see from the revised manuscript, the modifications suggested were included and the results better interpreted.
L53-54: If I well understand, the removal values provided refer to WWTP influent and primary effluent, or to secondary effluent or to tertiary effluent and not to the specific contribution of the step (primary, secondary and tertiary). I suggest to improve the clarity of the sentence.
Authors’ response: The Authors thank the Reviewer for the comment. The values reported in the introduction refer to the primary, secondary and tertiary effluents and not to the contribution provided by each specific step. The sentence was modified as follows: “Indeed, in a typical conventional activated sludge (CAS) based WWTP, the concentration of MPs after primary treatments is reduced up to 70%, whereas it further reduced from 80-90% and 99% in the secondary and tertiary effluents, respectively”.
L62-64: Ref 11 is a review and the results was not of the research group.
Authors’ response: The reference was replaced with the correct one.
L82-97: I suggest to add if the sewer to the three WWTP is mixed or separate and the type of users served by the WWTP (domestic, industrial, zootechnic…). These details could be of interest to better compare the results achieved for the three WWTPs. For instance, at lines 292 authors related the occurrence of some particles with to tires and road runoff.
The only provided detail is that the three WWTPs treat domestic wastewater and are of different capacity. My doubt is how the differences in shape and composition can be due to the wastewater type treated at the specific plant?
Authors’ response: The Authors thank the Reviewer for suggestion. All the WWTP were served by combined sewer systems. The following sentence was added in the revised manuscript: “All the WWTPs were served by combined sewer systems”.
This certainly affect the abundance of microplastics entering the WWTPs and the occurrence of tires particles. This aspect was also confirmed by previous studies. More discussion about this was added in the revised manuscript. Moreover, for a better comparison of the results with previous literature additional references were added.
Referring the last part of the comment, the Authors believe that is very difficult to correlate the microplastic shape and composition with the type of wastewater treated. Actually, a previous study reported that the composition of microplastics in the wastewater depends on the season, because of the living habits of local people. Based on this consideration, since all the samples were collected in the same period, it is reasonable to assess that the composition of microplastics was similar in all the plants. Instead, the treatment capacity of the plants could have affected the amount of microplastics detected in the sludge samples. Indeed, several studies reported that the amount of microplastics found in the sludge is proportional to the volume of wastewater treated (Van Do et al., 2022, https://doi.org/10.1016/j.eti.2022.102286). Based on this consideration, since the treatment capacity of the MBR (WWTP3) was much lower that the WWTP1, in theory the amount of microplastics detected in the sludge should have been lower. Instead, the fact that the microplastics detected in the MBR samples were higher although the lower treatment capacity suggested that for equal treatment capacity the amount of microplastics in the MBR sludge would be even higher. This confirmed that the amount of microplastics in the sludge of MBR is significantly higher than conventional WWTPs.
L95: Is it correct that W1 serves 330.000 PE and it has a capacity of 3600 m3/d? Check served population and capacity!
Authors’ response: The Authors thank the Reviewer for the comment and apologize for the mistake. The correct treatment capacity of W1 is 86,200 m3/d. The Authors also corrected the treatment capacity of W2 and W3.
Figure 2 enlarge axis titles and all the characters. It is not easy to evaluate the standard deviations in the bottom graph. As to pictures, it is not clear to which samples they refer to.
Authors’ response: The characters of the axis title were enlarged, and the colour of the graph changed to better highlight the standard deviation. The pictures are referred to microplastics detected in the MBR samples. The figure caption was modified to better clarify this.
L209: which are shapes could be present but you did not find?
Authors’ response: The only one shape that we did not find was the sphere. This could be attributed to the extraction methods used in this study that likely caused the spheres deterioration during the acidification steps. Additional comments about this were added in the revised version of the manuscript.
Figure 5: I recommend to add MP acronyms soon after their full name
Authors’ response: The figure was modified according to the Reviewer’s suggestion.
L293-294: what does this mean? At line 104 Authors wrote three dewatered sludge samples were taken in the three WWTPs in the same days (in April, May and June 2021) and at line 294, they wrote samples were taken after rain event. Is this collection constraint necessary and within the scope of the current study?
Authors’ response: The Authors apologize for the misunderstanding. Actually, the reference to rain events was referred to the findings reported in another study in which the authors found more tires particles after rain events. In the present study, no rain events occurred before each sampling. The high abundance of tires particles in the microplastics detected in this study was related to the sewage system that being of unitary type caused the entering of the first rain waters into the WWTPs.
General comments to figures: enlarge the characters.
Authors’ response: The figures were modified according to the Reviewer’s suggestion.

Reviewer 2 Report
Please find the comments on the attached document.

Author Response
Reviewer 2
The study examined the presence of Microplastics in the sludge of three wastewater treatment plants operated with different processes. Microplastics were detected in the sludge presenting an environmental pollution problem of microplastics into the environment.
Authors’ response: Authors thank the Reviewer#2 for the interesting suggestions useful for improving the quality of the paper. The Authors accepted all the suggestions and modified the manuscript accordingly to the Reviewer’s comments.
Abstract
Line 15 – write the full name for MPs as it is introduced for the first time in the article.
The rest of the abstract is well presented.
Authors’ response: Edit as suggested.
Keywords
Use a full name for MPs
Authors’ response: Edited according to the Reviewer’s comment.
Graphical abstract
The words on the graphical abstracts (CAS WWTP and MBR WWTP) are too small, making it difficult for the reader to read. Increase the font.
Authors’ response: The font of the words in the graphical abstract was increased according to the Reviewer’s comment.
Introduction
The introduction Is too brief. Authors should add more content on other technologies presented in literature for the recovery of MPs from wastes.
Authors’ response: The introduction was extended by adding more information about MPs removal using other technologies.
Methodology
Using hydrogen peroxide and sulphuric acid to treat MPs introduce another environmental problem. What happens to the effluent after treatment?
Authors’ response: The Authors thank the Reviewer for the comment. The use of hydrogen peroxide and sulphuric acid was necessary to remove all the residual organic matter from the microplastics extracted from the sludge. This procedure was adopted based on previous studies that dealt about the optimization of MPs separation from activated sludge. This is a laboratory method aimed at detecting the amount of MPs in a small sludge sample. Therefore, since the extraction procedure is carried out on a small sample, it involves the use of very small volume of these chemicals that are disposed according to the current regulations regarding the laboratory chemical wastes.
Results
The results section is well presented and below are minor corrections to be implemented:
Line 185 – should be, could also be attributed.
Authors’ response: Edit as suggested.
Line 186: delete ones.
Authors’ response: Edit as suggested.
Line 189 – should be, could be attributed.
Authors’ response: Edit as suggested.
Line 220 – rephrase the sentence.
Authors’ response: The sentence was rephrased as follows: “The greater percentage of films observed in the MBR sludge samples in comparison with the CAS was attributed to the retention effect exerted by the ultrafiltration membranes on these particles. Indeed, since film particles are bigger than the membrane porosity, they were entirely entrapped within the system and accumulated in the waste sludge.”
Line 221, 228 – attributed
Authors’ response: Edit as suggested.
Line 280 – full names for PET, CA, PB, PL
Authors’ response: Edit as suggested.
Line 283 – composed of not by
Authors’ response: Edit as suggested.
Line 300 – rephrase the sentence, greater diversity of plastics was observed.
Authors’ response: Edit as suggested.
Line 305 – as discussed above.
Authors’ response: Edit as suggested.
Line 322 – This “is” because
Authors’ response: Edit as suggested.
Line 341 – rephrase does not make sense
Authors’ response: The sentence was rephrased as follows: “The greater abundance of fibers and the greater diversity of plastics found in the MBR sludge, confirmed that MBR are more effective than CAS in removing MPs from wastewater. Nevertheless, the findings of this study confirmed also microplastics removed from wastewater are transferred in the waste sludge.”
Conclusion
The section is well presented, the author may consider recommending a solution for the MPs recovered from the sludge.
Authors’ response: The Authors added an additional consideration in the conclusion section for further studies to carry out about this topic.

Round 2
Reviewer 1 Report
The revised paper can now be considered for publication.